# Hemodynamic and Clinical Profiles of Pulmonary Arterial Hypertension Patients with GDF2 and BMPR2 Variants

**DOI:** 10.3390/ijms25052734

**Published:** 2024-02-27

**Authors:** Mei-Tzu Wang, Ken-Pen Weng, Sheng-Kai Chang, Wei-Chun Huang, Lee-Wei Chen

**Affiliations:** 1Institute of Emergency and Critical Care Medicine, National Yang Ming Chiao Tung University, Taipei 112, Taiwan; meitzuwang@gmail.com; 2Department of Critical Care Medicine, Kaohsiung Veterans General Hospital, Kaohsiung 813, Taiwan; 3Congenital Structural Heart Disease Center, Department of Pediatrics, Kaohsiung Veterans General Hospital, Kaohsiung 813, Taiwan; kenpenweng@gmail.com; 4Excelsior Biopharma Inc., Taipei 115, Taiwan; ryan.chang@excelsiorgroup.com.tw; 5Department of Medicine, School of Medicine, National Yang Ming Chiao Tung University, Taipei 112, Taiwan; 6Department of Physical Therapy, Fooyin University, Kaohsiung 813, Taiwan; 7Department of Surgery, Kaohsiung Veterans General Hospital, Kaohsiung 813, Taiwan; 8Department of Biological Sciences, National Sun Yat-Sen University, Kaohsiung 813, Taiwan

**Keywords:** BMPR2, GDF2, heritable pulmonary arterial hypertension, pulmonary arterial hypertension, whole exome sequencing

## Abstract

Asians have a higher carrier rate of pulmonary arterial hypertension (PAH)-related genetic variants than Caucasians do. This study aimed to identify PAH-related genetic variants using whole exome sequencing (WES) in Asian idiopathic and heritable PAH cohorts. A WES library was constructed, and candidate variants were further validated by polymerase chain reaction and Sanger sequencing in the PAH cohort. In a total of 69 patients, the highest incidence of variants was found in the BMPR2, *ATP13A3*, and *GDF2* genes. Regarding the *BMPR2* gene variants, there were two nonsense variants (c.994C>T, p. Arg332*; c.1750C>T, p. Arg584*), one missense variant (c.1478C>T, p. Thr493Ile), and one novel in-frame deletion variant (c.877_888del, p. Leu293_Ser296del). Regarding the *GDF2* variants, there was one likely pathogenic nonsense variant (c.259C>T, p. Gln87*) and two missense variants (c.1207G>A, p. Val403Ile; c.38T>C, p. Leu13Pro). The *BMPR2* and *GDF2* variant subgroups had worse hemodynamics. Moreover, the *GDF2* variant patients were younger and had a significantly lower GDF2 value (135.6 ± 36.2 pg/mL, *p* = 0.002) in comparison to the value in the non-*BMPR2*/non-*GDF2* mutant group (267.8 ± 185.8 pg/mL). The *BMPR2* variant carriers had worse hemodynamics compared to the patients with the non-*BMPR2*/non-*GDF2* mutant group. Moreover, there was a significantly lower GDF2 value in the *GDF2* variant carriers compared to the control group. *GDF2* may be a protective or corrected modifier in certain genetic backgrounds.

## 1. Introduction

Pulmonary hypertension (PH) is defined as a mean pulmonary artery pressure (mPAP) > 20 mmHg based on the 2022 European Society of Cardiology (ESC)/European Respiratory Society (ERS) guidelines [1,2]. It can be divided into five groups based on different mechanisms [1,2]. As PH progresses, right ventricular heart failure develops once the right ventricle fails to supply adequate cardiac output, and the death rate ranges from 4.5 to 12.3 per million population [3]. There are similar sex and age distributions of PH patients in the Han population compared to Western population-based studies; however, a lower 1-year survival rate for Chinese PH patients was reported in a previous registry, which could be attributed to the unique genetic variants of Asian PH patients [4,5].

Around 25–30% of idiopathic pulmonary arterial hypertension (PAH) patients have an underlying genetic cause and could be classified as heritable PAH patients. The 6th World Symposium on Pulmonary Hypertension disclosed a list of genes that have different levels of connection to PAH [6]. Among those PAH-related genes, heterozygous germline variants in the bone morphogenetic protein receptor type-2 gene (BMPR2) account for approximately 70–80% of heritable PAH and 10–20% of idiopathic PAH cases [7]. The transforming growth factor-β (TGF-β) superfamily includes two major branches, including the TGF-β–activin–nodal branch and the bone morphogenetic protein (BMP)–growth differentiation factor (GDF) branch [8]. Loss or dysfunction in the balance between the TGF-β–activin–nodal branch and the BMP-GDF branch have played a critical role in predisposition and disease progression in PAH [8].

Several studies have indicated that Asian PAH has a higher carrier rate of PAH-related genetic variants than Caucasian PAH [9,10,11]. However, reports using whole exome sequencing (WES) for identifying genetic variants from Southeastern Asian idiopathic and heritable PAH cohorts are still rare. A better understanding of the mechanisms by which BMPR2 variants define a subclass of patients with more severe disease is critical to improve our knowledge of PAH. This study aimed to use WES to identify PAH-related genetic variants and investigate their hemodynamic and clinical profiles in a Taiwanese idiopathic and heritable PAH cohort.

## 2. Results

A total of 69 patients were enrolled in this study. A flowchart of the study is illustrated in Figure 1. The basic characteristics of the patients are reported in Table 1. There was a female predominance, and the average age at diagnosis was 50 ± 20 years. The average six-minute walking distance (6MWD) was 332 ± 127 m, mPAP was 41 ± 16 mmHg, pulmonary arterial wedge pressure (PAWP) was 14 ± 4 mmHg, and pulmonary vascular resistance (PVR) was 8 ± 7 Wood units. The peak tricuspid regurgitation peak gradient was 52 ± 30 mmHg, and the CI was 2.7 ± 1.1 L/min/m^2^. Pulmonary artery saturation was 66 ± 12%. The N-terminal prohormone of BNP (NT-proBNP) was 1869 ± 2988 ng/L. Most patients were in WHO Fc III (63.8%), with 49.3% patients presenting slow progression and 13% patients reporting a rapid progression of symptoms.

Of the 69 idiopathic PAH patients, 36 distinct rare variants of 20 genes from 28 patients (28/69, 40.6%; *EIF2AK4* was not counted due to the obscured heterozygous contribution of *EIF2AK4* in PAH) were identified (Figure 1). Six rare variants were classified as likely pathogenic (LP) or pathogenic (P) according to the ACMG 2015 guidelines, and the remaining thirty variants were classified as variants of uncertain significance (VUS). Among all the candidate genes, *BMPR2*, *ATP13A3*, and *GDF2* had the highest incidence of variants (Table 2). Details of the VUS of the *EIF2AK4* gene and other likely benign variants are listed in Table 2 and Appendix A, respectively. In addition, a pie chart (Figure 2) was illustrated to explain the distribution of genetic variants in the idiopathic PAH cohort.

We found four *BMPR2* variants in five female PAH patients (5/69, 7.2%) in this study (Table 2). There were two nonsense variants (c.994C>T, p. Arg332*; c.1750C>T, p. Arg584*) and one missense variant (c.1478C>T, p.Thr493Ile) that has been reported previously [14,15,16], and there was also one novel in-frame deletion variant (c.877_888del, p.Leu293_Ser296del). However, we did not find any gross insertion, deletion, or copy number changes (CNVs) on *BMPR2*. Among the study subjects, two patients (A100593 and A100719) were mother and daughter, and they had family aggregates of heritable PAH carrying a *BMPR2* variant (c.1750C>T, p. Arg584*) and an *ATP13A3* splicing variant (c.970+1 G>A). The onset time of symptoms was 58 years for the mother and 39 years for the daughter. The mPAP values at diagnosis were 56 and 60 mmHg for the mother and daughter, respectively, and their PVR values at diagnosis were 35.0 and 10.2 Wood units, respectively. Both of them were at the intermediate risk level based on the 2022 ESC/ERS guidelines [2]. According to ACMG 2015 guidelines, all four *BMPR2* variants were classified as likely pathogenic or pathogenic, and the *ATP13A3* variant was classified as likely pathogenic. There were three patients carrying novel *GDF2* variants in the study cohort (3/69, 4.3%). One rare *GDF2* variant (c.259C>T, p. Gln87*) was a nonsense-type variant and classified as likely pathogenic (LP). Another two rare *GDF2* variants (c.1207G>A, p. Val403Ile; c.38T>C, p. Leu13Pro) belonged to the missense type and were classified as VUS. The allelic frequencies of all three variants were absent in the normal East Asian population database (gnomAD).

The patients were classified into three subgroups: *BMPR2* (B) variant carriers, *GDF2* (G) variant carriers, and the non-*BMPR2*/non-*GDF2* (N)-gene subgroup. A further analysis of the characteristics of the subgroups is listed in Table 3. In comparison to the patients in the N subgroup, the *BMPR2* variant carriers had higher mPAP (B: 66 ± 15; N: 38 ± 13 mmHg, *p* < 0.001), higher PVR (B: 22 ± 9; N: 15 ± 12 mmHg, *p* < 0.001), worse cardiac index values (B: 1.5 ± 0.5; N: 2.8 ± 1.1 L/min/m^2^, *p* = 0.030), and worse pulmonary artery saturation (B: 54 ± 13; N: 67 ± 11%, *p* = 0.045). Additionally, in comparison with the N-gene subgroup, the *GDF2* variant carriers were younger (G: 25 ± 13; N: 51 ± 20 years, *p* = 0.048), had higher mPAP (G: 66 ± 13; N: 38 ± 13 mmHg, *p* = 0.001), and had higher PVR (G: 15 ± 12; N: 7 ± 5 mmHg, *p* = 0.048).

Comparing GDF2 (BMP9) concentrations in the N-gene subgroup (267.8 ± 185.8 pg/mL, n = 16), insignificantly lower circulating levels of BMP9 were demonstrated in the subgroup of *BMPR2* variant carriers (162.4 ± 28.8 pg/mL, *p* = 0.079, n = 5), and there were significantly decreased circulating BMP9 values in the *GDF2* variant carrier group (135.6 ± 36.2 pg/mL, *p* = 0.002, n = 3) (Figure 3A). One *GDF2* nonsense variant (c.259C>T, p. Gln87*) carrier was an 18-year-old male with a significantly low serum BMP9 concentration of 97.0 pg/mL compared to two other patients with a missense variant (168.84 and 140.94 pg/mL). Serum BMP10 levels (Figure 3B) were significantly higher in the *GDF2* variant carriers (218.3 ± 113.7 pg/mL, n = 3) compared to the *BMPR2* variant carriers (105.3 ± 5.8 pg/mL, n = 5) and the N-gene subgroup (132.1 ± 59.2 pg/mL, n = 16).

The correlations regarding the hemodynamic/clinical data of the female *BMPR2* carriers vs. the N-gene subgroup (Appendix A) and male *GDF2*-carriers vs. the N-gene subgroup (Appendix A) are demonstrated in the Appendix A for the present study. In comparison to the N-gene subgroup, female *BMPR2* variant carriers (Appendix A) had higher mPAP (66 ± 15 vs. 37 ± 14 mmHg, *p* < 0.001), higher PVR (22 ± 9 vs. 7 ± 5 mmHg, *p* < 0.001), and worse cardiac index values (1.6 ± 0.5 vs. 2.7 ± 1.1 L/min/m^2^, *p* = 0.021). Additionally, male *GDF2* variant carriers (Appendix A) had higher mPAP (66 ± 13; N: 39 ± 12 mmHg, *p* = 0.002) and higher PVR (G: 15 ± 12; N: 6 ± 5 mmHg, *p* = 0.048) compared to the N-gene subgroup.

Our function analysis was established by reprogrammed peripheral blood mononuclear cells of one of the GDF2 variant carriers, and we differentiated them into endothelial cells to carry out the migration and angiogenesis functional assay. Stronger migration (Figure 4A) and angiogenesis functions (Figure 4B) were observed.

## 3. Discussion

PAH patients with different genetic factors present a variety of clinical manifestations and outcomes. In the present study, the average age at diagnosis was 50 ± 20 years, with a female predominance. Compared with the patients in the N-gene subgroup, the *BMPR2* variant carriers had higher mPAP, higher PVR, worse CI, and worse pulmonary artery saturation. Moreover, younger ages, higher mPAP values, and higher PVR values have been reported in *GDF2* variant carriers.

### 3.1. Clinical Manifestations of PAH Patients with BMPR2 Variants

It has been established that genetic variation in the transforming growth factor β (TGF-β) signaling pathway plays an important role in the pathogenesis of idiopathic and heritable PAH, such as BMPR2, activin receptor-like kinase 1 (ACVRL1), and endoglin (ENG) [6,17,18,19]. Abnormalities in *BMPR2* result in the aberrant activation of the TGF-β signaling pathway and the regulation of cell growth or apoptosis of pulmonary artery vascular smooth muscle cells [20]. Patients bearing *BMPR2* genetic variants are usually younger and have higher mPAP or PVR at diagnosis [7,21,22]. The present study reported five female patients with *BMPR2* variants. Although male patients have been reported to have a higher *BMPR2* mutation rate compared to female patients [23], the female patients had higher penetrance than the males, with 42% and 14%, respectively [4,24,25]. Moreover, it has been shown that being female is the single most definitive modifier for the penetrance of *BMPR2* mutations in PAH [6,25]. The present study reported 5 females with *BMPR2* variants (Appendix A), which could be explained by the superior penetrance of *BMPR2* mutations in female patients compared to male patients.

Furthermore, genotype–phenotype correlation analyses have demonstrated that *BMPR2* variant patients were younger at diagnosis (27.2 vs. 31.6 years, *p* = 0.0003) compared with those without *BMPR2* mutations [26]. However, the present study reported that a younger age was insignificant in the *BMPR2* mutation subgroup in comparison to the N-gene subgroup (43 ± 11 vs. 51 ± 20 years, *p* = 0.626, Table 3). This result was limited by the small sample size, comprising only five patients. Furthermore, *BMPR2* variant carriers had a significantly shorter time to lung transplantation (approximately 10 years earlier than non-carriers) [7,9,11,22]. This study demonstrated that BMPR2 variant carriers had a higher mPAP, higher PVR, worse CI, and worse pulmonary artery saturation (Table 1), which is consistent with the poor outcomes of patients with *BMPR2* mutations in previous studies.

### 3.2. Potential Mechanisms of the GDF2 Variant and PAH

GDF2 (BMP9) is a ligand of the *BMPR2* receptor. There are rare coding variants of *GDF2*, which occurred in 6.7% of cases in this study and has been implicated as a causative gene for the development of PAH [10]. The first case report of a 5-year-old boy with homozygous *GDF2* nonsense variants had severe PAH and right heart failure at 3 years of age [27]. More studies have indicated that the involvement of *GDF2* in PAH is facilitated through autosomal dominant behavior because there are heterozygous variants in *GDF2* [10,19,28,29,30]. In addition to the *BMPR2* and *GDF2* variants, *ACVRL1* and *ENG* variants also encode receptors belonging to the TGF-β superfamily and affect vascular proliferation in a manner similar to *BMPR2* [19,31,32]. Furthermore, more downstream genes of the TGF-β/BMP pathway, associated with the BMP axis, have been reported, such as *SMAD1/5/8*, *SMAD4*, and *BMPR1B* [33].

Unlike other GDFs, the GDF2 ligand binds to the endothelium-specific BMPR2/activin receptor-like kinase 1 (ALK1) heterotetrametric complex, activates intracellular phosphorylation relay through the BMP-specific SMAD1/5/8 signaling route to ensure the antiproliferative ability, and downregulates the proliferative effect via the activin receptor type IIA (ActRIIA)–SMAD2/3 pathway [34,35]. In patients with the *GDF2* variant, a decreased expression of the GDF2 ligand might prompt GDF2 to bind to ActR-IIA/IIB instead of BMPR2, which drives a similar impact to the BMPR2 variant on endothelial cell proliferation [36]. Sotatercept is a novel drug for PAH therapy that has the ability to sequester excess ActRIIA ligands and thereby rebalance between two counterbalancing signaling pathways [34]. Also, an in vivo study demonstrated that GDF2 suppressed endothelial cell proliferation in healthy control subjects but increased it in PAH patients with BMPR2 variants, which supported evidence of switching to the ActR-IIA/IIB rather than the BMPR2 downstream signaling pathway in PAH [36].

A previous study demonstrated that the *BMPR2* variant can lead to endothelial dysfunction and PAH in the absence of protective modifiers [37]. For example, the overexpression of endoglin increases the GDF2 response, which is opposite to the result of silencing BMPR2 or ActRIIA expression [35]. Moreover, another study indicated that daily GDF2 administration had no significant effect on mice bearing a heterozygous endothelial *BMPR2* deletion (*BMPR2*^EC+/−^) but led to excessive angiogenesis in *BMPR2^EC^*^−/−^ mice, which implied that the impact of the *GDF2* variant on PAH was associated with the *BMPR2* variant [36]. The present study demonstrated an extremely low serum GDF2 (BMP9) concentration in a *GDF2* nonsense variant (c.259C>T, p.Gln87*), which supports the mechanism of PAH caused by the loss of normal signaling via the BMP9-BMPR2-SMAD1/5/8 signaling route. However, whether *GDF2* is a protective or corrected modifier in certain genetic backgrounds [36,38,39,40] or whether the *GDF2* variant leads to PAH by overexpressing the ActR-IIA/IIB-SMAD2/3 signaling route warrants further investigation [36].

All three *GDF2* variant carriers in the present study were male. Regarding the *GDF2* mutation, a investigation into gender disparity was obscured. There were similar hemodynamic appearances between the male *GDF2* variant carriers and general PAH cohort in our subanalysis (Appendix A), but the influence of younger age at onset in the *GDF2* variant carriers was absent due to our small sample size. However, gender’s contribution in *GDF2* mutation-associated PAH requires further investigation. The subanalysis of the present study illustrated that gender’s contribution was not remarkable in both *BMPR2* and *GDF2* variant carriers (Appendix A).

With regard to the interaction between GDF2 (BMP9) and BMP10, circulating BMP9 was encoded by the *GDF2* gene [41], and one study indicated that despite there being no differences in plasma BMP10 levels in male PAH patients compared to a control group, the plasma BMP10 levels in female PAH patients were significantly lower than the control females [42]. In our study, BMP10 had a counterbalancing-like elevated level compared to the reduced GDF2 (BMP9) level (Figure 3B). Whether there was counterbalancing protein interaction between the two productions of GDF2 encoding and whether the character was contributed to a certain *GDF2* variant or the whole picture of the *GDF2* mutation warrants further investigation.

### 3.3. Genetic Variants Other Than BMPR2 and GDF2 in PAH Patients

Due to inadequate knowledge of the entire PAH gene background in Asian populations, the present study did not include patients who were not specifically targeted for “screened variants” as a control group. Instead, the present study took patients without *BMPR2* and *GDF2* (N-gene subgroup) variants as a more convincing control group.

In addition to genes directly affecting the TGF-β/BMP pathway, this study reported four *NOTCH* variant carriers. NOTCH receptor signaling controls smooth muscle cell proliferation and maintains smooth muscle cells in an undifferentiated state, which has been reported to be associated with the pathogenesis of PAH and is involved in pulmonary vascular morphogenesis [43]. *NOTCH3* KO mice demonstrated a protective effect against hypoxia-induced PAH, and a γ-secretase inhibitor that inhibits the NOTCH pathway could be applied in the treatment of PAH in a mouse model [44]. However, only a few genetic variants of the NOTCH pathway in PAH patients have been reported [45,46].

*ATP13A3* was also identified in the present study. *ATP13A3* encodes a P-type ATPase, a cation channel transporter, and has been reported to be associated with PAH [6,19,47]. The monoallelic *ATP13A3* variant has been associated with adult-onset PAH [2,4,5,20], and Machado et al. demonstrated that biallelic *ATP13A3* variants in childhood-onset PAH are characterized by extreme morbidity and mortality [48]. The PAH patients with or without *ATP13A3* genetic variants are reported in Appendix A, which illustrates that there were higher mPAP values (52 ± 7 vs. 38 ± 13 mmHg, *p* = 0.043) in the *ATP13A3* variant subgroup compared to the non-ATP13A3 variant carriers. In addition, two out of the four *ATP13A3* variant carriers also carried a nonsense mutation in the BMPR2 gene (A100593 and A100719), which demonstrated the speculative contribution of the two *ATP13A3* variants to PAH instead of the observed effect despite the small sample size. The two patients (A100593 and A100719) in the present study were mother and daughter, and they had family aggregates of heritable PAH, carrying a *BMPR2* variant (c.1750C>T, p. Arg584*) and an *ATP13A3* splicing variant (c.970+1 G>A). In this case, heterozygous variants in *ATP13A3* could act as a second hit for the development of PAH; thus, they require further investigation. Due to the study design adopting the process of “retrospective enrollment” and “prospective screening”, the two subjects were diagnosed as idiopathic PAH at initial diagnosis. However, we realized they should have been classified as heritable PAH after a genetic investigation.

The detection of biallelic pathogenic *EIF2AK4* mutations is able to make precise and accurate molecular diagnoses of autosomal recessively inherited pulmonary capillary hemangiomatosis (PCH) and pulmonary veno-occlusive disease (PVOD), and this was documented in the 6th World Symposium on Pulmonary Hypertension [6]. A carrier of biallelic *EIF2AK4* counts for up to 25% of patients with sporadic PVOD or PCH [6]. However, *EIF2AK4* can also play a “second hit” role, contributing to autosomal dominantly inherited hereditary PAH [49]. The present study separated the variants into VUS and likely benign variants to facilitate further investigations on the heterozygous contribution between *EIF2AK4* and PAH in the future. Furthermore, the PAH patients with or without *EIF2AK4* genetic variants are reported in Appendix A. However, there was no significant difference between the *EIF2AK4* variant and control subgroups.

Higher carrier rates of some PAH-related genes in Asian populations compared to Caucasian populations have been reported. In this study, *BMPR2* variants accounted for around 15% in idiopathic PAH and more than 50% in heritable PAH. The present study reports five patients with BMPR2 variants (7.2%). Moreover, the *GDF2* variants are relatively common (6.7%) in Chinese PAH patients [10], and three patients bearing *GDF2* variants (4.3%) were reported in this study. Furthermore, the *PTGIS* gene was reported as a causative gene specifically in Chinese patients in [50], and four PAH patients (5.8%) in this study had variants of this gene. The findings regarding variants that were obtained in the present study align with those of previous publications. However, the carrier rates are inconsistent with previous studies due to the small sample size of this study.

### 3.4. Limitations and Strengths

Previous studies have investigated gene variants in Asian PAH patients, including in Chinese pediatric cohorts and Korean cohorts [9,11]. A recent study in central Taiwan identified 14 potential PAH-related genetic variants by using WES analysis to study 45 PAH patients [51]. However, these studies lacked data on *GDF2* variants, and there is no functional assay to validate the contribution of *GDF2* variants in Asian cohorts. In contrast, *BMPR2* has been relatively well studied compared to other gene variants, and a key novel aspect of this study was the fact that it placed more focus on *GDF2* and compared the clinical appearances of *GDF2* and *BMPR2* variants and carried out the migration and angiogenesis functional assay (Figure 4).

The purpose of this study was to investigate PAH-related genetic variants in Asian idiopathic and heritable PAH cohorts using sequencing and filtering strategies such as the two-tiered approach. The use of exploratory data for this study’s Asian cohort was limited by the inadequate genetic database for the PAH cohorts. In spite of this limitation, the present study could still provide insights into the puzzle of how ethnic differences and genetic background affect PAH. In addition, the threshold of 0.0001 in all control/healthy individuals with no higher allele frequency in the subgroups is a more conservative threshold and in line with PAH incidence. However, the greater the number of novel variants with lower minor allele frequency that are present in a sample population, the more difficult it will be to home in on the causal gene [52]. Furthermore, A MAF cutoff of 0.001 has previously been recommended for filtering variants responsible for dominant Mendelian disorders [52]. Due to the inadequate sample size of the present study and the limited genetic background data in the gnomAD database and Taiwan Biobank, a minor allele frequency of 0.001 was used to define variants as very rare variants, and a cutoff of 0.001 was adopted in the present study [53].

In spite of its small sample size, this study used WES to identify PAH-related genetic variants and to investigate their hemodynamics, clinical profiles, and functional assay results in Asian idiopathic and heritable PAH cohorts. Novel genetic variants were detected in an Asian database of PAH cohorts. Further validation is required to confirm the pathogenic variants likely associated with PAH.

## 4. Materials and Methods

The cohort of idiopathic PAH patients was “retrospectively enrolled”, with the earliest diagnostic date of this cohort being 4th March 2001 and the most recent diagnostic date being 28 March 2022. However, for our investigation into genetic background, patients were “prospectively screened”, starting from 1 August 2021 to 31 December 2023. In addition, the demographics, clinical presentations, hemodynamics, and multi-parameters of risk assessment of patients diagnosed with PAH were documented for analysis. However, the lack of available survival data represents a limitation of this study. The Institutional Review Board of Kaohsiung Veterans General Hospital approved this study (KSVGH21-CT8-11). Written informed consent was obtained from all the participants.

### 4.1. Serum GDF2 (BMP9) and BMP10 of PAH Patients

Serum GDF2 concentrations were measured using the Human BMP-9 DuoSet^®^ RRELISA kits (R&D Systems, Inc., Catalog number: DY3209, Minneapolis, MN, USA) according to the manufacturer’s instructions. GDF2 levels were analyzed in patients with *BMPR2* and *GDF2* or non-*BMPR2* and non-*GDF2* variants. The concentrations of BMP10 in the culture medium or patient serum was measured using enzyme-linked immunosorbent assay kits (DuoSet R&D Systems, Catalog number: DY2926-05) according to the manufacturer’s instructions.

### 4.2. Hemodynamics and Cardiopulmonary Function Tests

Pulmonary hypertension was defined as a mean pulmonary arterial pressure >20 mmHg at rest, as assessed by right heart catheterization according to international guidelines [2]. The World Health Organization (WHO) functional class (Fc), 6MWD, peak oxygen consumption, ventilatory equivalents for carbon dioxide (VE/VCO2), NT-proBNP, heart rate, right atrial pressure, cardiac output (CO), cardiac index (CI), pulmonary artery saturation, pressure, mPAP, PAWP, PVR, and peak tricuspid regurgitation peak gradient were analyzed using hemodynamic and cardiopulmonary function tests.

### 4.3. DNA Extraction, Whole Exome Sequencing, and Data Analysis

Genomic DNA was isolated from the peripheral blood of all patients using a NucleoSpin^®^ Blood Kit (Macherey-Nagel, Duren, Germany). Whole exome sequencing was conducted using the KAPA HyperExome Plus Kit (Roche Sequencing and Life Science, Indianapolis, IN, USA) according to the KAPA HyperCap Workflow v3.0. The WES libraries were sequenced using paired-end 2 × 150 bp on a NovaSeq 6000 Sequencer (Illumina Inc., San Diego, CA, USA). Raw fastq data were processed using the DRAGEN platform (SW:05.021.595.3.7.5) (Illumina). The coverage of specific genes was analyzed using WES Binary Alignment/Map (BAM) files. In this study, the average coverage of all the candidate genes was greater than 100×. Variants with a read depth less than 30× were filtered out. All quality control-passed variants from the Variant Call Format (VCF) files were further screened using a list of 39 PAH-related genes, including *BMPR2* [10], *GDF2* [10], *ATP13A3* [10], *AQP1* [10], *ACVRL1* [10], *KCNA3* [54], *KCNK3* [10], *EIF2AK4* [10], *KLF2* [10], *SMAD1* [10], *SMAD4* [10], *SMAD9* [10], *TBX4* [10], *SOX17* [10], *CAV1* [10], *KCNA5* [10], BMPR1A [30], *BMPR1B* [30], *ENG* [6], *ABCA3* [9,55], *NOTCH1* [56,57], *NOTCH2* [58,59], *NOTCH3* [46,60], *ABCC8* [61,62], *BMP10* [42,63], *FBLN2* [64], *JAG1* [65], JAG2, *PTGIS* [50], *PDGFD* [64], *GGCX* [66], *SMAD5* [67,68], *KLK1* [66], *SARS2* [69], *SERPINE1* [61], *SIRT3* [70], *THBS1* [71], *TNIP2* [72], and *TopBP1* [57,73]. Normal subject control databases (gnomAD database: 807,162 samples with 2.78% East Asian and Taiwan Biobank: 189,107 samples) were used to exclude gene variants with allelic frequencies greater than 0.1% in the East Asian group (EAS). The impacts of all variants on specific genes were analyzed using Polyphen2 (http://genetics.bwh.harvard.edu/pph2/, access on 2 December 2023) and SIFT software (http://blocks.fhcrc.org/sift/SIFT.htmlhttp://genetics.bwh.harvard.edu/pph2/, access on 2 December 2023), and clinical evaluations were performed according to the American College of Medical Genetics and Genomics (ACMG) 2015 guidelines [74]. Furthermore, we applied ExomeDepth from a free R package (version 1.1.15) to screen copy number changes or structural variations in the *BMPR2* gene in all PAH patients [75].

### 4.4. Validation of Variants by Polymerase Chain Reaction and Sanger Sequencing

The candidate variants were validated by polymerase chain reaction (PCR) and Sanger sequencing. Briefly, specific PCR primers were designed manually, and PCR was conducted using PCRBIO Ultra Polymerase (PCR Biosystems, London, UK) and SensoQuest Labcycler (SensoQuest, Göttingen, Germany). Purified PCR products were sequenced using an ABI PRISM Terminator Cycle Sequencing kit v3.1 on an ABI 3730 DNA sequencer (Applied Biosystems, Bedford, MA, USA). Validation data for candidate variants, obtained using PCR and Sanger sequencing, are shown in Appendix A.

### 4.5. Generation of PAH-Specific iPS Cell Lines and Assessment of Function Analysis

Seven individuals were recruited into this study, including three patients diagnosed with idiopathic PAH, one *GDF2* variant carrier (A100537), and three healthy individuals with no history of cardiovascular disease (Ctrl). There was no significant bias in gender or age in the three subgroups. The primary cultures of peripheral blood mononuclear cells were established from all individuals and were used for reprogramming at passages 2–4. Sendai virus delivery of OCT4, KLF4, c-Myc, and SOX2 was used to generate 5–8 independent iPSC lines per individual. These iPSC cell lines were thoroughly characterized and shown to be fully reprogrammed to pluripotency, as evaluated by colony morphology, growth dynamics, sustained long-term passaging (>20 passages), alkaline phosphatase (AP) staining (Figure 5A), the expression of pluripotency-associated transcription factors (OCT4, SOX2, and NANOG) and surface markers (SSEA4 and TRA1-60) (Figure 5B), the silencing of Sendai virus transgenes (Figure 5C), in vitro pluripotent differentiation ability and the generation of teratomas comprising derivatives of the three main embryo germ layers (Figure 5D), and karyotype stability (Figure 5E). Moreover, sequence analysis of the *GDF2* gene in the iGDF2, iPAH, and control–iPS cell clones confirmed the heterozygous *GDF2* mutation in the iGDF2-iPS cell clone (Figure 5F). Three replicates with the same patient were established in each subgroup. The peripheral blood mononuclear cells of one of the *GDF2* variant carriers were reprogrammed and differentiated into endothelial cells. Further migration and angiogenesis functional assays were established by using endothelial cells (Figure 4).

### 4.6. Generation and Characterization of iPSC-Derived Patient-Specific Endothelial Progenitor Cells (EPCs) and Endothelial Cells (ECs)

A number of studies have shown that CD34+ circulating progenitor cells (CPCs) and several subsets of CD34+ cells (such as CD133+/KDR+) can participate in vascular repair and growth and may be associated with vascular endothelial function. Therefore, these cells may reflect vascular integrity and have been used as biomarkers of vascular repair. CD34+ CPCs are a diverse group of progenitors, consisting of both hematopoietic and non-hematopoietic CPCs, with CD133 and KDR often used as more definitive antigen markers for endothelial progenitor cells (EPCs) [28]. Friedrich et al. demonstrated that CD34−/CD133+/VEGFR-2+ progenitor cells are precursors of CD34+/CD133+/VEGFR-2+ EPCs and functionally more potent than CD34+/CD133+/VEGFR-2+ EPCs with respect to homing and vascular repair [19]. In this study, we have established a procedure for the direct differentiation of human iPSCs to mesoderm-lineage cells via a chemically defined monolayer differentiation protocol. As shown in Figure 6A, human iPSCs can be differentiated to endothelial progenitors by 3 days of culture in an optimized defined medium supplemented with GSK3 inhibitors and another 7 days in StemPro-34 medium. On day 10 of differentiation, the differentiated cells developed typical EPC appearance, with an initial spindle shape and expressed EPC markers (CD133 and KDR). They could form a typical cobblestone-like shape of endothelial cells and express endothelial markers, including CD31 and VE-cadherin (CD144), through to day 21 post-differentiation. On day 10 of induction differentiation, FACS analysis showed that around 95% and 75% of the cell population expressed the EPC makers CD133 and KDR, respectively, and 1% of cells expressed CD34, indicating that the differentiated cells on day 10 were apparently precursors of “classical” CD34+/133+ EPCs. On day 21 of differentiation, around 75% and 99% of the cells stained positive for the EC maker CD31 and CD144, respectively, and 49% of cells expressed CD34, indicating that the CD34−/133+/KDR+ cells on day 21 could give rise to an endothelial progeny (Figure 6B). No differences were observed in the differentiation efficiency of iPSCs from the control, iPAH, and iGDF2 individuals. The functional features of the endothelial cells from the control, iPAH, and iGDF2 individuals were also confirmed through the uptake of acetylated LDL and lectin binding by direct fluorescent staining (Figure 6C), indicating that the iPSC-ECs from all subjects showed similar EC cobblestone morphology, acetylated LDL uptake, and UEA-1 lectin binding.

### 4.7. Statistical Analysis

In this study, all continuous variables are expressed as mean ± standard deviation (SD), and categorical variables are expressed as numbers and percentages. The Kolmogorov–Smirnov test was applied to check the normality of normal distributions. For clinical analysis, the 69 PAH patients were categorized into three groups: *BMPR2* genetic variant carriers, *GDF2* genetic variant carriers, and a N-gene subgroup. Continuous variables were compared using a one-way ANOVA with Tukey’s post hoc test. Categorical variables were examined using the chi-square test and Fisher’s exact test. The serum GDF2 concentrations between the subgroups of non-*BMPR2*/*non-GDF2*, *BMPR2*, and *GDF2* were analyzed using the Kruskal–Wallis test. Statistical significance was set at a two-tailed *p*-value < 0.05. The SPSS 19 statistical software package (SPSS, Inc., Chicago, IL, USA) was used for all calculations.

## 5. Conclusions

This study investigated WES-based genetic variants in an Asian PAH cohort. The most common variants were BMPR2 (c.994C>T, p. Arg332*; c.1750C>T, p. Arg584*; c.1478C>T, p. Thr493Ile; c.877_888del, p. Leu293_Ser296del), ATP13A3, and GDF2 (c.259C>T, p. Gln87*; c.1207G>A, p. Val403Ile; and c.38T>C, p. Leu13Pro). Compared with the patients in the N-gene subgroup, the BMPR2 variant carriers had higher mPAP, higher PVR, worse CI, and worse pulmonary artery saturation. The GDF2 variant carriers in the PAH cohort had worse clinical and hemodynamic performances and were younger in age. Moreover, there was a significantly lower GDF2 value in the subgroup of GDF2 variants than in the non-mutant subgroup. Whether GDF2 is a protective or a corrective modifier in certain genetic backgrounds requires further investigation.

## Figures and Tables

**Figure 1 ijms-25-02734-f001:**
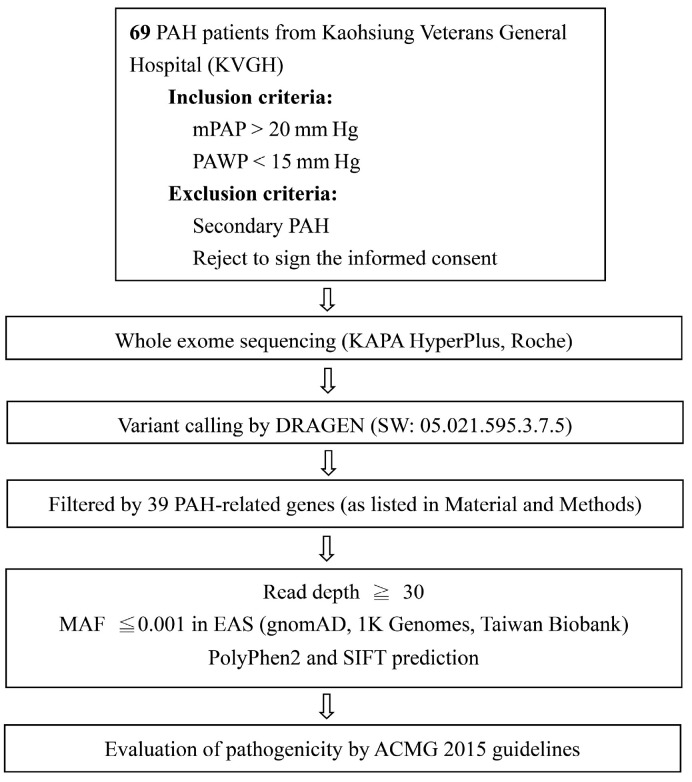
Study flowchart.

**Figure 2 ijms-25-02734-f002:**
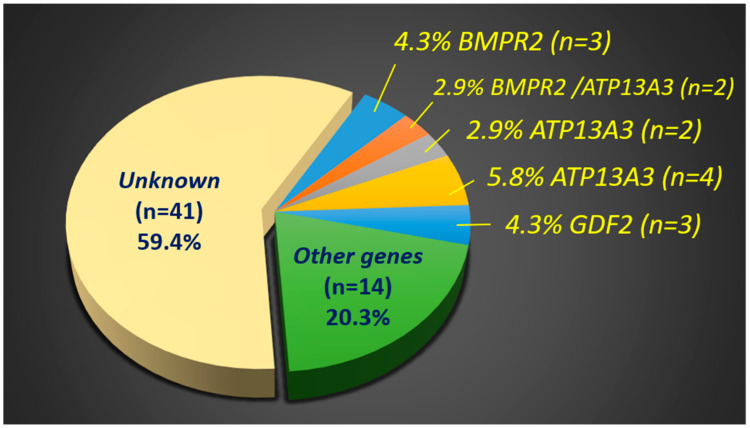
Pie chart of the distribution of the genetic variants of the idiopathic PAH cohort.

**Figure 3 ijms-25-02734-f003:**
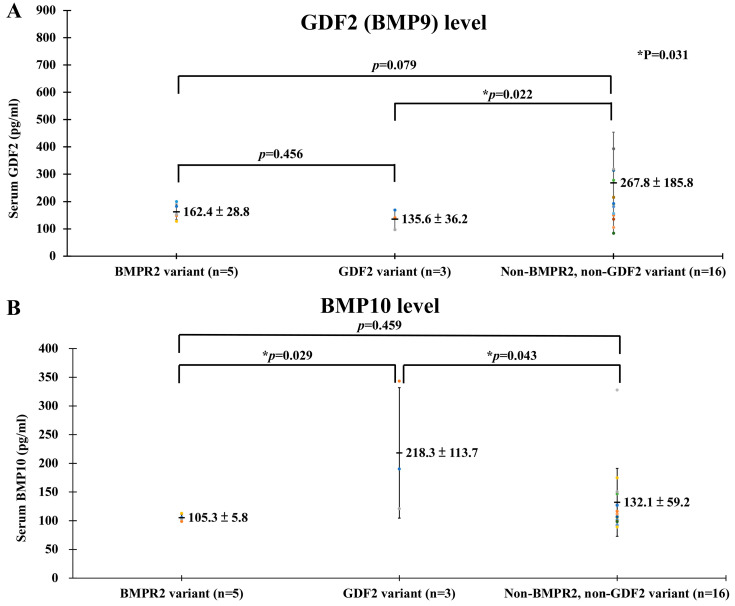
Serum GDF2 (BMP9) and BMP10 concentrations in subgroups of *BMPR2* variant carriers, *GDF2* variant carriers, and the N-gene subgroup. Values are mean +/− SD; n = number of patients in subgroup. Panel (**A**): insignificantly lower circulating levels of BMP9 were demonstrated in the *BMPR2* variant carriers, and significantly lower BMP9 levels were reported for the *GDF2* variant carriers compared to the GDF2 concentrations in the N-gene subgroup. Panel (**B**): serum BMP10 levels were significantly higher than in the *GDF2* variant carriers compared to the *BMPR2* variant carriers and the N-gene subgroup. (* means statistical significance).

**Figure 4 ijms-25-02734-f004:**
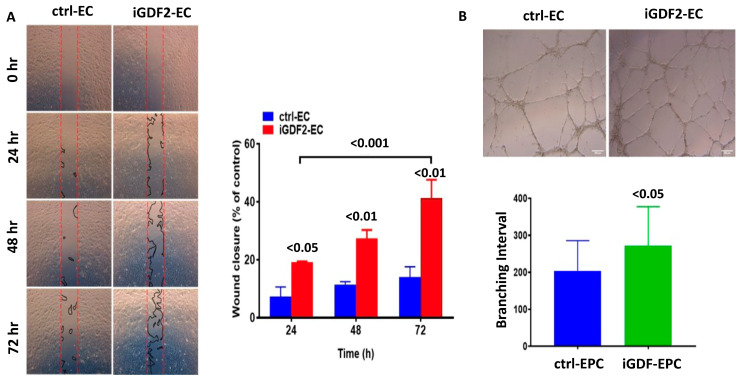
Function analysis was carried out by reprogramming the peripheral blood mononuclear cells of one of the GDF2 variant carriers and differentiating them into endothelial cells to conduct the migration and angiogenesis functional assay. Stronger migration (**A**) and angiogenesis functions (**B**) were observed.

**Figure 5 ijms-25-02734-f005:**
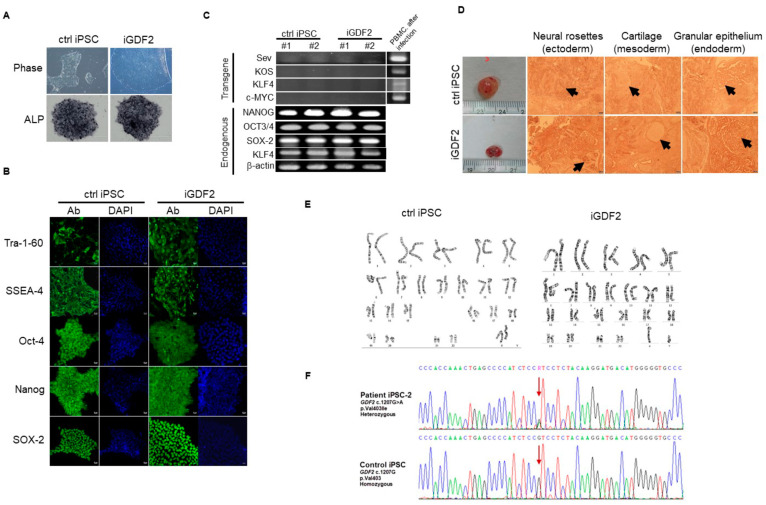
Characterizations of cells fully reprogrammed to pluripotency in three iPSC cell lines. Panel (**A**): colony morphology, growth dynamics, sustained long-term passaging (>20 passages), alkaline phosphatase (AP) staining. Panel (**B**): expression of pluripotency-associated transcription factors (OCT4, SOX2, and NANOG) and surface markers (SSEA4 and TRA1-60). Panel (**C**–**E**): silencing of Sendai virus transgenes (**C**), in vitro pluripotent differentiation ability and generation of teratomas comprising derivatives of the three main embryo germ layers (**D**), and karyotype stability (**E**). Panel (**F**): sequence analysis of the GDF2 gene in the iGDF2, iPAH, and control–iPS cell clones confirmed the heterozygous GDF2 mutation in the iGDF2-iPS cell clone.

**Figure 6 ijms-25-02734-f006:**
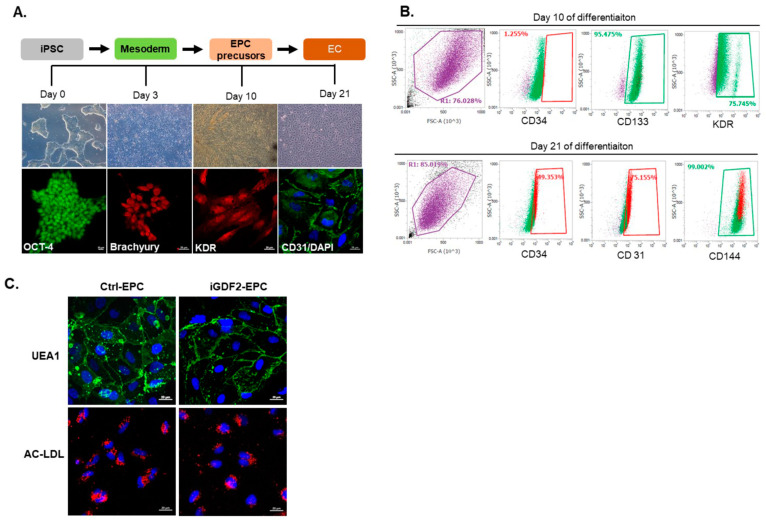
Generation and characterization of iPSC-derived patient-specific endothelial progenitor cells (EPCs) and endothelial cells (ECs). Panel (**A**): Human iPSCs differentiated into mesoderm on D3 by being cultured in an optimized defined medium supplemented with GSK3 inhibitors. It took another 7 days in StemPro-34 medium for the cells to develop a typical EPC appearance, along with an initial spindle shape and expressed EPC markers (CD133 and KDR). On D21, the ECs with typical cobblestone-like shapes expressed endothelial markers, including CD31 and VE-cadherin (CD144). Panel (**B**): FACS analysis on D10 showed that the EPC maker of CD133 was expressed in 95% of the cell population, and KDR was expressed in 75% of the cell population. In addition, CD34 was expressed in 1% of the cells. Both markers indicated that the differentiated cells on D10 are apparently precursors of “classical” CD34+/133+ EPCs. On D21 of differentiation, 75% and 99% of the cells stained positive for the EC makers CD31 and CD144, respectively, and 49% of cells expressed CD34, indicating that the CD34−/133+/KDR+ cells on day 21 could give rise to an endothelial progeny. Panel (**C**): There were no differences in the differentiation efficiency of the iPSCs between the control, iPAH, and iGDF2 individuals. The functional features of the endothelial cells from the control, iPAH, and iGDF2 individuals were also confirmed through the uptake of acetylated LDL and lectin binding by direct fluorescent staining, indicating that iPSC-ECs from all subjects showed similar EC cobblestone morphology, acetylated LDL uptake, and UEA-1 lectin binding.

**Table 1 ijms-25-02734-t001:** Basic characteristics of pulmonary arterial hypertension patients in Kaohsiung Veterans General Hospital (n = 69) †.

Characteristics		
Female (n, %)		51 (73.9)
Age at diagnosis, years		50 ± 20
Six-minute walking distance (m)		332 ± 127
Mean pulmonary arterial pressure (mmHg)		41 ± 16
Pulmonary arterial wedge pressure (mmHg)		14 ± 4
Pulmonary vascular resistance (Wood units)		8 ± 7
Peak tricuspid regurgitation peak gradient (mmHg)		52 ± 30
Peak oxygen consumption (mL/min/kg)		12 ± 4
Ventilatory equivalents for carbon dioxide (VE/VCO2)		39 ± 14
Right atrial pressure (mmHg)		13 ± 10
Cardiac index (L/min/m^2^)		3 ± 1.1
Pulmonary artery saturation (%)		66 ± 12
N-terminal prohormone of brain natriuretic peptide (ng/L)		1869 ± 2988
World Health Organization functional class (N, %)	I	4 (5.8)
	II	19 (27.5)
	III	44 (63.8)
	IV	2 (2.9)
Progression of symptoms	No	26 (37.7)
	Slow	34 (49.3)
	Rapid	9 (13.0)

† Data of continuous variables are expressed as mean ± SD; changes in categorical variables were analyzed by chi-square tests and are expressed as n (%).

**Table 2 ijms-25-02734-t002:** Details of genetic variants in PAH-related genes.

No.	ID	Sex	Gene	cDNA	Amino Acid	Variant	Genotype	PolyPhen2/SIFT	ACMG 2015 *	MAF in EAS
**Category A. PAH patients with *BMPR2* variants (N = 5)**										
**1**	A100553	F	*BMPR2*	c.877_888del	p. Leu293_Ser296del	Inframe_del	Het	N/A ($)	LP	0.00000
**2**	A100593	F	*BMPR2*	c.1750C>T	p. Arg584*	Nonsense	Het	N/A ($) [12]	P	0.00000
			*ATP13A3*	c.970+1G>A		Splicing	Het	N/A ($)	LP	0.00009
			*ABCA3*	c.635G>A	p. Arg212Gln	Missense	Het	PD/T	VUS	0.00022
**3**	A100655	F	*BMPR2*	c.994C>T	p. Arg332*	Nonsense	Het	N/A ($) [13]	P	0.00000
**4**	A100719	F	*BMPR2*	c.1750C>T	p. Arg584*	Nonsense	Het	N/A ($) [12]	P	0.00000
			*ATP13A3*	c.970+1G>A		Splicing	Het	N/A ($)	LP	0.00000
**5**	A110118	F	*BMPR2*	c.1478C>T	p. Thr493Ile	Missense	Het	PD/D [14]	LP	0.00000
			*ENG*	c.278G>A	p. Arg93Gln	Missense	Het	B/T	VUS	0.00019
**Category B. PAH patients with *GDF2* variants (N = 3)**										
**6**	A100537	M	*GDF2*	c.1207G>A	p. Val403Ile	Missense	Het	-/T ($)	VUS	0.00000
**7**	A100692	M	*GDF2*	c.38T>C	p. Leu13Pro	Missense	Het	-/D ($)	VUS	0.00000
			*BMP10*	c.475G>A	p. Asp159Asn	Missense	Het	B/T	VUS	0.00019
**8**	A110160	M	*GDF2*	c.259C>T	p. Gln87*	Nonsense	Het	N/A ($)	LP	0.00000
**Category C. PAH patients with variants of other PAH-related genes (N = 20)**										
**9**	A100538	F	*PTGIS*	c.592G>A	p. Val198Ile	Missense	Het	B/T	VUS	0.00038
**10**	A100540	M	*ATP13A3*	c.260G>A	p. Arg87His	Missense	Het	B/T	VUS	0.00000
**11**	A100554	F	*ATP13A3*	c.3040A>T	p. Ile1014Phe	Missense	Het	B/D	VUS	0.00000
**12**	A100559	F	*KLK1*	c.119C>T	p. Ala40Val	Missense	Het	B/T	VUS	0.00019
**13**	A100588	F	*SIRT3*	c.415A>G	p. Arg139Gly	Missense	Het	PD/D	VUS	0.00019
**14**	A100591	F	*NOTCH2*	c.7177G>C	p. Ala2393Pro	Missense	Het	PD/T	VUS	0.00000
**15**	A100607	F	*THBS1*	c.3272G>A	p. Arg1091His	Missense	Het	PD/D	VUS	0.00019
			*KCNA3*	c.283G>A	p. Ala95Thr	Missense	Het	B/T	VUS	0.00034
**16**	A100641	F	*PTGIS*	c.635G>A	p. Arg212Gln	Missense	Het	B/T	VUS	0.00019
**17**	A100643	M	*TNIP2*	c.248C>G	p. Ser83Trp	Missense	Het	-/D ($)	VUS	0.00019
**18**	A100653	F	*BMPR1B*	c.664A>G	p. Lys222Glu	Missense	Het	PD/T	VUS	0.00000
			*NOTCH1*	c.3549G>T	p. Glu1183Asp	Missense	Het	B/T	VUS	0.00019
			*PTGIS*	c.592G>A	p. Val198Ile	Missense	Het	B/T	VUS	0.00038
**19**	A100654	F	*JAG2*	c.320C>G	p. Pro107Arg	Missense	Het	PD/T	VUS	0.00059
**20**	A100683	F	*JAG2*	c.359G>C	p. Arg120Pro	Missense	Het	PD/D	VUS	0.00063
			*NOTCH1*	c.3973G>A	p. Ala1325Thr	Missense	Het	PD/T	VUS	0.00019
**21**	A100697	F	*SMAD4*	c.554C>T	p. Pro185Leu	Missense	Het	PD/D	VUS	0.00019
			*NOTCH3*	c.67C>T	p. Pro23Ser	Missense	Het	B/T	VUS	0.00000
**22**	A100710	F	*TOPBP1*	c.1366A>G	p. Lys456Glu	Missense	Het	N/A ($)	VUS	0.00000
**23**	A100720	M	*JAG1*	c.3286C>T	p. Arg1096Trp	Missense	Het	PD/D	VUS	0.00000
**24**	A100740	F	*TOPBP1*	c.75A>T	p. Lys25Asn	Missense	Het	B/D	VUS	0.00000
**25**	A110041	F	*ABCA3*	c.143C>T	p. Ser48Leu	Missense	Het	PD/T	VUS	0.00058
**26**	A110136	F	*TNIP2*	c.281T>C	p. Ile94Thr	Missense	Het	PD/T	VUS	0.00019
**27**	A110138	F	*PTGIS*	c.860A>G	p. Asn287Ser	Missense	Het	PD/T	VUS	0.00039
**28**	A110161	M	*KLK1*	c.469G>A	p. Gly157Ser	Missense	Het	PD/D	VUS	0.00000

Abbreviation: F, female; M, male. Reference sequences: BMPR2 (NM_001204.7), ATP13A3 (NM_001367549.1), ABCA3 (NM_001089.3), ENG (NM_000118.3), GDF2 (NM_016204.4), BMP10 (NM_014482.3), PTGIS (NM_000961.4), KLK1 (NM_002257.4), SIRT3 (NM_001370310.1), NOTCH2 (NM_001200001.2), KCNA3 (NM_002232.5), THBS1 (NM_003246.4), TNIP2 (NM_001161527.2), BMPR1B (NM_001203.3), NOTCH1 (NM_017617.5), JAG2 (NM_002226.5), NOTCH3 (NM_000435.3), SMAD4 (NM_005359.6), TOPBP1 (NM_001363889.2) and JAG1 (NM_000214.3). PolyPhen2: PD, Probably damaging; D, Damaging; B, Benign. SIFT: D, Deleterious, T, Tolerated. N/A or minus symbol (−), not applicable. * ACMG 2015: The American College of Medical Genetics and Genomics guidelines: P, pathogenic; LP, Likely pathogenic; VUS, variant of uncertain significance. MAF in EAS: Minor allele frequency of East Asian in gnomAD exome databases. ($) PolyPhen2 and SIFT in silico prediction program were unable to analyze frameshift, nonsense, in frame deletion, gross deletion types and some variants with unknown reasons.

**Table 3 ijms-25-02734-t003:** Pulmonary arterial hypertension patients with or without the genetic variants bone morphogenetic protein receptor type 2 (*BMPR2*) and growth differentiation factor 2 (*GDF2*), as well as clinical and hemodynamic presentations at initial diagnosis (n = 69) †.

Characteristics		*BMPR2* (n = 5)	*GDF2* (n = 3)	N-Gene (n = 61)	*p*-Value	
					*BMPR2* vs. N-Genet	*GDF2* vs. N-Gene
Female (N, %)		5 (100)	0 (0)	46 (75.4)		
Age of onset, years		43 ± 11	25 ± 13	51 ± 20	0.626	0.048
Six-minute walking distance (m)		351 ± 165	490 ± 101	322 ± 122	0.851	0.051
Mean pulmonary arterial pressure (mmHg)		66 ± 15	66 ± 13	38 ± 13	<0.001	0.001
Pulmonary arterial wedge pressure (mmHg)		11 ± 3	11 ± 3	14 ± 5	0.299	0.499
Pulmonary vascular resistance (woods)		22 ± 9	15 ± 12	7 ± 5	<0.001	0.048
Peak tricuspid regurgitation peak gradient (mmHg)		64 ± 35	58 ± 23	50 ± 31	0.573	0.903
Peak oxygen consumption (mL/min/kg)		12 ± 4	15 ± 6	12 ± 4	0.977	0.501
Ventilatory equivalents for carbon dioxide (VE/VCO2)		50 ± 18	45 ± 4	38 ± 14	0.148	0.754
Right atrial pressure (mmHg)		14 ± 9	11 ± 5	13 ± 10	0.995	0.902
Cardiac index (L/min/m^2^)		1.6 ± 0.5	2.6 ± 1.1	2.8 ± 1.1	0.030	0.966
Pulmonary artery saturation (%)		54 ± 13	68 ± 10	67 ± 11	0.045	0.956
N-terminal prohormone of brain natriuretic peptide (ng/L)		4117 ± 5336	228 ± 46	1763 ± 2763	0.172	0.616
World Health Organization functional class (N, %)	I	1 (20)	0 (0)	3 (4.9)	0.469	0.389
	II	1 (20)	2 (66.7)	16 (26.2)		
	III	3 (60)	1 (33.3)	40 (65.6)		
	IV	0 (0)	0 (0)	2 (3.3)		
Progression of symptoms	No	1 (20)	1 (33.3)	24 (39.3)	0.417	0.523
	Slow	4 (80)	1 (33.3)	29 (47.5)		
	Rapid	0 (0)	1 (33.3)	8 (13.1)		

† Data of continuous variables are expressed as mean ± SD; changes in categorical variables were analyzed by chi-square tests and are expressed as n (%).

## Data Availability

Data are unavailable due to privacy or ethical restrictions.

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
