# Peer review of "Hemodynamic and Clinical Profiles of Pulmonary Arterial Hypertension Patients with GDF2 and BMPR2 Variants"

_ijms, 2024, doi:10.3390/ijms25052734_

Round 1

Reviewer 1 Report (New Reviewer)

Comments and Suggestions for Authors

Wang and colleagues performed whole exome sequencing in a cohort of 69 PAH patients aiming at identifying PAH-related genetic variants in the Asian population. Despite the small sample size and the lack of survival data, the paper provides some insights on novel variants that could potentially play a role in PAH.

Major comments:

1. In the abstract (lines 27-28) and in the method section (lines 481-487) the authors describe further validation of candidate variants using PCR and Sanger sequencing, however the data are not shown. The validation should be added in the supplementary material with relevant information regarding number of variants/patients tested, detailed materials and methods used, and sequences of all the primers should be provided.

2.    In the abstract (lines 35-37) the authors state that “GDF2 variant patients were younger and had a significantly lower GDF2 value (135.6± 36.2 pg/mL, P=0.002), in comparison to the value in the non-mutant subgroup (267.8 ± 185.8 pg/mL)”. However, it is my understanding that the patients referred as “non-mutant” do have mutations in other PAH-related genes (as shown in Table 3). For this reason, defining them as “non-mutant” is misleading.

3.    Have some of the patients recruited in this study already been analysed in Wang MT et. al (Glob Heart. 2021; Ref. [4])? If so, that should be stated in the manuscript.

4.    In Figure 3, the number of patients in each subgroup should be stated. Also, “non- mutant patients” (line 206) in figure legend should be changed to non-BMPR2/non-GDF2 mutant patients (as for comment 1). The figure legend should describe the figure (i.e., values are mean +/- SD, n=, and statistical test used), not explain the results (the text is indeed a repetition of lines 177-185).

5.    In the method section the description of how peripheral blood mononuclear cells from one of the GDF2 variant carriers (which variant?) were differentiated into endothelial cells and detailed methods for migration and angiogenesis assays (used to generate Figure 4) are missing - please add. Endothelial cells markers should also be shown to confirm differentiation. In Figure 4, the number of replicates should be stated (i.e. three independent experiments?). Please also change the legend for Figure 4 as in point 4 (lines 226-228) and add p-values.

6.    In the discussion section (lines 392-395) it is stated that mPAP is higher in patients with ATP13A3 genetic variants compared to “controls”. It is my understanding that the authors here compared PAH patients with and without ATP13A3 genetic variants within the same cohort of patients, hence the “control group” includes patients with all the other variants. If that is correct, I would clarify it in the discussion. Moreover, 2 out of the 4 patients carrying ATP13A3 genetic variants also carry a nonsense mutation in the BMPR2 gene (A100593 and A100719), making it virtually impossible to separate the individual contribution of the two variants to the observed effect. For this reason, and given the small sample size, this finding seems a little speculative, and it should be taken with caution.

7.    The discussion section could be better written as it feels very disconnected. I strongly suggest reviewing it.

8.    I personally agree with the choice of using non-BMPR2/non-GDF2 mutant patients for the comparison in the manuscript (discussion section, lines 376-379). However, referring to this group as a “control group” feels a bit misleading. I would suggest the use of N-gene subgroup (as in methods line 169) throughout the paper. Besides, this should be mentioned earlier on in the discussion.

Minor comments:

1.    All the acronyms in the manuscript should be described the first time they are used (i.e. Results section, lines 82-85, MWD, mPAP, etc.). Acronyms used in Figure 1 should be explained in the figure legend as well.

2.    The font used in Figure 3 is quite small and it is not clear what “*P=0.031” (Panel A, top right) and “P=0.071” (panel B, top right) refer to. That should be stated clearly in the figure legend.

3.    In Table 3 it is not clear what some symbols used mean (i.e., $).

4.    The statistical test to check for normal distribution should be mentioned in the methods (lines 506-515).

5.    The analysis carried out separating male and female patients (lines 361-373) should be moved to the result section and only discussed in the discussion.

6.    The informed consent statement and study number should be added at the beginning of the method section (lines 446-452) instead of at the end of the manuscript (lines 543-547).

7.    Please consider reordering the paragraphs in materials and methods (i.e., function tests should be after patient enrolment).

Comments on the Quality of English Language

I would strongly suggest reviewing the English of the manuscript.  The discussion section in particular needs rewriting.

Author Response

Reviewer 2 Report (New Reviewer)

Comments and Suggestions for Authors

The authors report the results of genetic screening for pathogenic (or likely pathogenic) variants in pulmonary arterial hypertension genes using an Asian cohort. The focus on the Asian population is relevant as there is a push in the genetic community for greater genetic diversity and better representation of non-European populations. However, the report could be improved with more stringent filtering of the data and reorganization of the tables and figures.

Major comments

1. Abstract/Introduction/Results - The argument for greater genetic burden among Asian vs Western PAH cases is weakened by the relatively small samples sizes of the Asian cohorts. It is not likely that greater genetic burden will be supported by the current dataset after more conservative filtering of the genes and variants.

2. Methods/Results - The inclusion of 39 genes in the screen for PAH-causative variants is too broad as many of the genes are not well-supported in the literature. The authors could consider a 2-tiered approach, including well-supported genes in tier 1 and less-supported candidate genes in tier 2 (exploratory data in the Asian cohort). However, most of the variants identified in less-supported would be excluded with more conservative variant filtering.

3. Methods/Results - The allele frequency threshold of 0.001 in EAS is higher than the disease frequency in the population. Further, the incidence of PAH has no known genetic ancestry bias. A threshold of 0.0001 in all control/healthy individuals with no higher AF in sub-groups is a more conservative threshold and in line with PAH incidence.

4. Methods/Results - The use of in silico pathogenicity prediction tools like CADD or REVEL are recommended over single metric tools like SIFT and polyPHEN. Thresholds of CADD >20 and REVEL >0.75 are widely-used and highly recommended.

5. Figure 1 - This is a flow chart and should not include results without details. Recommend deleting the last two rows of boxes (i.e. the rows beneath ACMG classification).

6. Table 1 - The variant table should be presented before Figures 3 and 4. The data should be presented by gene, in alphabetical order, rather than by arbitrary categories. Variants in the same gene should be listed by location,  5' to 3'. Recommend deleting the first column and the genotype column (all het and this could be mentioned in the title of the table), replacing polyPHEN/SIFT with CADD or REVEL, moving ACMG classification to be the last column, adding AF gnomAD CTLs_all and using the complimentary gnomAD CTLs_EAS as the next column. For gnomAD or similar database, the version and population used should be specified (i.e. gnomAD v2.1.1). Only include variants that meet the AF and in silico prediction thresholds.

7. Figure 3 - Increase the font size to increase readability. The Y-axes labels should indicate that "serum levels" or "circulating levels" of BMP9/10 are being measured. Provide the n for each group. Insignificant findings should be described as "no difference," rather than "lower but insignificant."

8. Figure. 4 - The photos are too dark for easy interpretation. Increase the font size. Add the n for each group and the statistical test performed to the legend.

Minor

1. Change "GDF2 values" to circulating levels of BMP9" or similar throughout.

Comments on the Quality of English Language

The quality of English language should be improved. This includes word choices and verb tenses.

Author Response

Reviewer 3 Report (New Reviewer)

Comments and Suggestions for Authors

The manuscript titled 'Hemodynamic and Clinical Profiles of Pulmonary Arterial Hypertension Patients with GDF2 and BMPR2 Variants' investigates the hemodynamic and clinical profiles of patients with pulmonary arterial hypertension who have GDF2 and BMPR2 variants. The study's objective is to identify genetic variants related to PAH by using whole exome sequencing in an Asian cohort with idiopathic and heritable PAH. The manuscript presents a well-conducted study that provides a comprehensive analysis of genetic variants and their association with hemodynamics, offering valuable insights into understanding PAH in the Asian population.

The study addresses a gap in the literature by focusing on GDF2 variants in addition to BMPR2 in Asian PAH patients. Identifying new genetic variants and their potential impact on the progression of pulmonary arterial hypertension (PAH) contributes to the existing knowledge in the field.

Suggestions for improvement: 

The material and methods section does not describe the assessment of function analysis by reprogramming peripheral blood mononuclear cells of one of the GDF2 variant carriers and their differentiation into endothelial cells to evaluate migration and angiogenesis functional assay. 

Some sentences are complex and may benefit from simplification for better clarity. Additionally, when introducing abbreviations such as MWD, mPAP, PAWP, PVR, and CI, their complete forms should be provided upon first mention to enhance reader understanding (lines 82-86).

Although the discussion provides a comprehensive overview, the authors should discuss potential limitations and avenues for future research. Addressing the limitation of a small sample size and its impact on generalizability could further strengthen the discussion.

Figures can effectively convey information. However, providing more detailed captions for each figure, explaining the abbreviation in figure legends, and discussing their specific relevance in the text would enhance their utility.

The manuscript is a valuable contribution to the understanding of PAH in the Asian population, particularly in relation to GDF2 variants. Addressing the suggestions would further enhance the manuscript's quality and impact.

Comments on the Quality of English Language

Some sentences are complex and may benefit from simplification for better clarity. 

Round 2

Reviewer 3 Report (New Reviewer)

Comments and Suggestions for Authors

The authors have significantly improved the manuscript, described their methods, and added research limitations. The authors have addressed all these points, and the manuscript is suitable for publication.

This manuscript is a resubmission of an earlier submission. The following is a list of the peer review reports and author responses from that submission.

Round 1

Reviewer 1 Report

Comments and Suggestions for Authors

This is potentially a very interesting manuscript aiming to find PAH-related genetic variants in Southeastern Asian PAH patients. In this study by Wang et al., titled “Hemodynamic and Clinical Profiles of Pulmonary Arterial Hypertension Patients with GDF2 and BMPR2 Variants”, the authors used whole exome sequencing to correlate gene variants with hemodynamic and clinical profiles in idiopathic and heritable PAH patients from Taiwan. The study focuses on BMPR2 and GDF2 gene mutations. The data indicate that both gene variant carriers have worse hemodynamic and clinical parameters while GDF2 variant carriers also were younger than subjects in non-BMPR2 or non-GDF2 variant group. Furthermore, GDF2 serum level was significantly lower in GDF2 variant group. Based on these data the authors concluded that GDF2 may be a protective or corrected modifier in certain genetic backgrounds. However, overall the authors are encouraged to add novelty to the manuscript by including new experimental data. I have some concerns about the selection of control subjects which diminish my enthusiasm. The manuscript would also benefit from the discussion on how the current findings are aligned with the published reports PAH-related genetic variants in Asian vs Caucasian PAH subjects.

Major comments:

1.      What is the novelty of the study? BMPR2 mutations have been previously reported in Asian PAH patients including Taiwan PAH patients (please see PMID: 35811711; ref#9, 11). GDF2 mutations in PAH patients have also been published (PMID: 31661308; PMID: 29650961).

2.      Based on mean±SD provided in Table 3 it seems like you have a perfect normal distribution in GDF2 mutations group with small sample size (n=3) while typically larger sample size (n≥30) is considered sufficient to hold a perfect normal distribution where the mean, median, and mode are the same value. Please consider presenting the data as Box-Whisker or scatter plots to better allow for reader appreciation of variability within groups. This also applies to Fig. 2.

3.      It seems that all 61 patients (including 20 patients with other PAH-related gene variants from Table 2) were included in non-BMPR2/non-GDF2 subgroup in Table 3 and Fig. 2. Interesting that only 28 patients were included in Table 2. Do other 41 patients out of total 69 patients have no mutations for any 39 genes screened? If so, those 41 patients would be more appropriate control group (instead of all 61 patients) for Table 3 and Fig. 2 to exclude the effect of other PAH-related genes on hemodynamic/clinical data and GDF2 level in 20 patients.

4.      BMPR2 variants were found in females only while GDF2 variants were found in males only (Table 3). Are there any correlations in hemodynamic/clinical data between BMPR2 carriers (females) vs non-BMPR2/non-GDF2 carriers (females) or between GDF2-carriers (males) vs non-BMPR2/non-GDF2 carriers (males)?

5.      What is the rationale to provide EIF2AK4 genetic variants in a Suppl. Table S1 rather than include it in Table 2? There is no hemodynamic/clinical data and discussion on patients with EIF2AK4 mutations while n=9. It would also be interesting to see hemodynamic/clinical data on ATP13A3 variants carriers since the authors listed this gene as one with the highest incidence of variants (n=4) in PAH cohort.

Minor concerns:

1.      Throughout the manuscript the authors state that BMPR2 and GDF2 variant subgroups are younger and have worse hemodynamics however, data show that GDF2 subgroup but not BMPR2 subgroup has a significant difference in age compared with non-carriers. Please revise accordingly. Given into the consideration the title, the conclusion on BMPR2 variants should be added to the abstract. Interesting that published reports demonstrated the younger age of BMPR2 mutations patients (PMID: 29743074; PMID: 35811711). Would the authors be able to speculate on discrepancies in Discussion section?

2.      It would be helpful for unfamiliar readers to indicate that in humans GDF2 encodes the circulating BMP9 and keep the labels consistent (please see Fig. 2).

3.      What is the timeline of the study? Please consider to add the survival data in Table 3.

4.      Are your findings aligned with the published literature indicating the unique genetic variants and/or higher carrier rate of PAH-related genes in Asian vs Caucasian population?

5.      On Page 1/Line 81 it states “Of 69 idiopathic PAH patients…” while on Page 2/Line 122 it is mentioned that two subjects (mother and daughter) have a heritable PAH. Please clarify PAH type for patients.

6.      Based on the Materials and Methods Section total 39 PAH-related genes were screened in this study. The data on 20 genes are listed in Table 2. Are the authors able to provide the data on other 19 genes?

7.      The manuscript would benefit from proof-reading. For example, Table 1 states “Filtered by 37 PAH-related genes (listed in Material and Methods)” however Materials and Methods indicate 39 PAH-related genes. Fig. 2 legend and Page 3/Line 173 have some internal comments which probably should be removed. The values from Table 3 are cited incorrectly in the text such as on Page3/Line 164. On Page 8/Line 261 it states “…this study reported four Notch3 variant carriers…” however only one Notch3 carrier is listed in Table 2.

8.      On Page 3/Lines 169-171 it states “Comparing the GDF2 concentrations in the non- BMPR2/non-GDF2 subgroup (267.8±185.8 pg/mL), lower GDF2 values were demonstrated in subgroups of BMPR2 (162.4±28.8 pg/mL, P=0.079), and GDF2 (135.6± 36.2 pg/mL, P=0.002) (Fig. 2).” Please revise the sentence since the difference in GDF2 concentrations between non-BMPR2/non-GDF2 and BMPR2 subgroups are not significant based on P value.

9.      The authors specifically emphasize that GDF2 nonsense variant carrier (18-year-old male) has a very low GDF2 serum level but it is unclear what speculation is made.

10.  On Page 8/Lines 293-298 the references citations are not consistent with the rest of the manuscript and confuse the reader (i.e. NOTCH115,16; ABCA310,14 etc).  

Author Response

Dear Reviewer:

The authors are very much thankful to the reviewers for their thorough review and comments on the revised article. Here, we further revised this paper in the light of reviewers’ important suggestions and comments. We hope the revision has improved the paper to a level of their satisfaction. Number wise answers to the specific comments are as follows. Please see the attachment for the revisions.

Thank you for your consideration. We look forward to hearing from you.

Sincerely,

Wei-Chun Huang, MD, PhD; E-mail: wchuanglulu@gmail.com

Department of Critical Care Medicine, Kaohsiung Veterans General Hospital, No. 386,

Lee-Wei Chen, MD, PhD; Email: lwchen@vghks.gov.tw

Department of Surgery, Kaohsiung Veterans General Hospital, Kaohsiung, No. 386, Dazhong 1st Rd., Zuoying Dist., Kaohsiung City 813, Taiwan

Tel: 886-7-3468278; Fax: 886-7-3455045

Reviewer 2 Report

Comments and Suggestions for Authors

This study by Wang et al., aimed to identify pulmonary arterial hypertension (PAH)-related genetic variants using whole exome sequencing in a Taiwanese PAH cohort. The authors sequence a total of 69 patients and find the highest incidence of variants in BMPR2, ATP13A3, and GDF2, with the BMPR2 and GDF2 variant subgroups showing worse hemodynamics. As expected by other publications, the GDF2 variant subgroup had significantly lower GDF2 values.

The article is well written, has a clear objective, and succeeds at reporting novel mutational findings. It also raises an interesting point by highlighting the differences between the mutational load in Asian PAH patients when compared with most of the big studies (mostly Caucasian populations within the United States and Europe).

I understand that functional studies were out of scope for this article, but I would encourage the authors to collaborate with other scientists to reduce the number of Variants of Uncertain Significance reported in the future.

I do not have any technical concerns about the data reported, but I have several minor points that could be improved in the manuscript.

Minor points:

·        Out of 69 patients, the authors were able to find rare variants in up to 28 of them, which is quite high, but only 6 of them are classified as Likely Pathogenic or Pathogenic. I suggest the authors use the percentage of cases explained by genetics also (which is quite similar to what is found in other populations, with around 15% of the IPAH cases being explained by genetics).

·        Why are the EIF2AK4 cases depicted in a different table? I understand this gene lacks functional studies and a clear mechanism in PAH, but it can be reported with the rest.

·        The case of the A100593 and A100719 family is interesting, as the ATP13A3 variant could act as a second hit for the development of PAH. If the authors agree maybe this could be highlighted. Also, the ATP13A3 variant is a very good candidate for a minigene assay in case they want to validate it in the future.

·        I would encourage the authors to make a chart showing the % of mutations per gene, there are many examples in the literature, and something like a pie chart would be very informative, it would be helpful to highlight the differences in percentage with other studies in the discussion.

·        The finding of the lower levels of BMP9 in their BMPR2 mutants is interesting, but the lower amount in patients with variants in GDF2 was expected and reported (PMID:31661308, which also showed a way to carry functional analyses for this gene, is not mentioned in the discussion or anywhere). Also, Figure 2 should show the individual values in all the groups. The levels of BMP10 would be something worth checking in the future.

·        I would advise complementing the tables with plots showing the most interesting findings as well as the individual values and the statistics, it would highly improve the readability of the article. There are free tools online like ggplotter that could help.

·        In line 230 I am not sure that is the right citation if the authors want to speak about disease causative mutations reported, as that one is a classical article showing the importance of the BMP axis. Also, there is an erratum in 229 “SAMD” instead of “SMAD”.

·        I recommend the authors check the latest curation of PAH genes (PMID:37422716).

·        There are several reports of NOTCH3 mutations that the authors could check (PMID: 24936512, 34199176).

·        Check lines 293-297 there seem to be either citations missing or random numbers on the list of genes in the panel.

Although I understand that sometimes making the data public is not possible, I would encourage the authors to upload the VCF files into a repository, their data could be invaluable for the research of PAH in understudied populations. Also, there is value in the common variation that could be explored in the future.

Comments on the Quality of English Language

The article does not need English editing. The text is concise and well-written. 

Author Response

Dear Reviewer:

The authors are very much thankful to the reviewers for their thorough review and comments on the revised article. Here, we further revised this paper in the light of reviewers’ important suggestions and comments. We hope the revision has improved the paper to a level of their satisfaction. Number wise answers to the specific comments are as follows.Please see the attachment for the revisions.

Thank you for your consideration. We look forward to hearing from you.

Sincerely,

Wei-Chun Huang, MD, PhD; E-mail: wchuanglulu@gmail.com

Department of Critical Care Medicine, Kaohsiung Veterans General Hospital, No. 386,

Lee-Wei Chen, MD, PhD; Email: lwchen@vghks.gov.tw

Department of Surgery, Kaohsiung Veterans General Hospital, Kaohsiung, No. 386, Dazhong 1st Rd., Zuoying Dist., Kaohsiung City 813, Taiwan

Tel: 886-7-3468278; Fax: 886-7-3455045

Round 2

Reviewer 1 Report

Comments and Suggestions for Authors

The major comments were not addressed very well.

 Major comments:

1.      The novelty is still somewhat low since the functional assays are not included in the study.

2.      There is no revision for Table 3. The pie chart is not helpful.

3.      Ok.

4.      There is no discussion on Suppl. Tables S2 and S3 however there is significance in some hemodynamic parameters. 18 patients were listed in Suppl. Table S3 but the table heading refers to N=69.

5.      There is no revision on EIF2AK4. There is no revision on ATP13A3. Four patients with ATP13A3 mutations were listed in Table 2 (ID #A100593, A100719, A100540, and A100554).

Author Response

Thanks for your comments. Please see the attatchment.
